# Combining Tabular and Satellite-Based Datasets to Better Understand Cropland Change

Kenneth Lee Copenhaver

CropGrower LLC, Tampa, FL 33606, USA; kencopenhaver@gmail.com

**Abstract:** In recent years, regulatory agencies in the USA and Europe have begun to require documentation that land used to produce crops and biofuels has not been converted from carbon-capturing grasslands or forests. Precise measurement of these land cover changes, however, has proven difficult. Analysis to date has focused primarily on moderate resolution (30 m) satellite imagery, which has not provided the land cover granularity or accuracy needed. These studies have estimated large-scale land conversion to crops in the USA. This study analyzed the satellite datasets but included tabular datasets and aerial imagery of the USA to determine whether the combination of datasets, focusing on more detailed analysis in these locations, could more accurately identify potential locations of land use change. Analyses of satellite imagery data from 1985 to 2020 found that much of the land that 2008 to 2020 satellite datasets classified as natural-to-crop land change was idle cropland. The results indicate a dynamic landscape of marginal land moving in and out of cropland. Approximately as much land was allowed to go fallow (6145 hectares) as land going into crop (7901 hectares) from 1985 to 2020. The results from this study indicate regulatory agencies could more accurately measure the impacts of conversion of natural lands to crop if long-term historical land cover/land use was also analyzed.

**Keywords:** land cover change; cropland change; satellite imagery

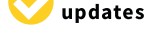



## 1. Introduction

Recent regulations in the USA and the European Union have required suppliers of biofuels to quantify conversion of natural lands to cropland to allow the purchase or subsidization of these products [1–3]. The agricultural landscape is vast, and quantification of land change is complex, making it difficult to identify these small changes [4]. Many previous studies that have measured change to cropland have focused on moderate resolution satellite imagery from one recent year to another (30 m) and often estimate very large changes from grassland to crop [5–7] Recent research, however, has found satellite imagery at the resolution used by these studies often overestimates change to cropland [8–10]. Other studies have combined moderate resolution satellite imagery with high resolution airborne imagery to estimate conversion to cropland but have found contradictory results based on their interpretation of the imagery. One study interpreted fallow cropland as crop [11] and estimated little change, while the other interpreted fallow or idle cropland as natural and measured greater change [12]. These studies, however, did not take long-term land use into account. Cropland in the USA declined from 1985 to 2006 [13], and crop and grassland have been relatively stable since 2001 [4]. This land, previously cropped, would be the most likely to be converted. If governments seek to protect natural lands, and companies are penalized for changes to cropland, the data used should be as accurate as possible. The goal of this study was to combine the moderate resolution satellite datasets with historical information across different data sources (satellite, tabular, and aerial imagery) to more accurately identify the amount and nature of long-term cropland change. While other studies have used the three different types of data in different combinations, this study will

use all three to look at cropland transitions over a 45-year period (1985 to 2020) to see if more specific answers can be discerned on land used for new cropland.

### 1.1. Measurement of Land Use Change

Understanding change from natural lands to cropland and vice versa requires a historical perspective. Year to year changes in total hectares in crops in the United States Department of Agriculture's (USDA) National Agricultural Statistics Service (NASS) crop acreage reports indicate that crop fields are not growing the same crops year after year [14]. Instead, the crop footprint is dynamic, with fields going in and out of production while planting different crops due to changes in commodity prices, input prices, crop rotation patterns, local demand, and weather [15]. Overall, land being used for crops in the USA has been trending downward since the 1970s; however, increased demand may be slowing the downward trend and even increasing the use of old agricultural lands or natural lands for new production (Figure 1).

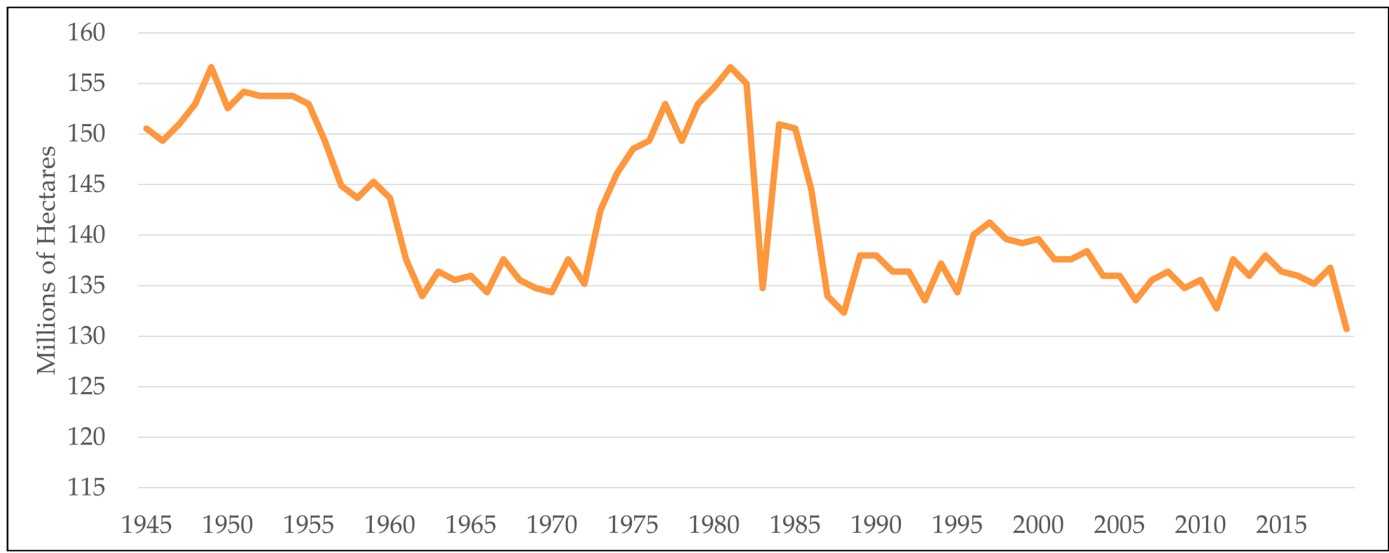

**Figure 1.** Land in Crops in the USA from 1945 to 2019 according to the USDA Major Land Uses Dataset.

Land that may have recently come into crop production could: (1) have been sitting idle for several years; (2) be part of the Conservation Reserve Program (CRP), which is land once in crop but now held in reserve funded by the USDA; (3) have been used in the past as intermittent hay or pasture; or (4) be coming from native grassland (native sod), which would release long-stored carbon reserves into the atmosphere, contributing to climate change [4].

The growing trend in cropland is mostly in the Midwest and Great Plains, with cropland declining in the eastern and western states (Figure 2). Since cropland in some Great Plains and Midwestern states has increased since 2007, it is possible that some natural grassland is being converted to crop. It is also possible that land in pasture, hay, or idle could be coming back into production.

According to language in the US Agricultural Act (2014 Farm Bill), native sod is land where "plant cover is composed principally of native grasses, grass like plants, forbs, or shrubs suitable for grazing and browsing;" and has "never been tilled, or the producer cannot substantiate that the ground has ever been tilled, for the production of an annual crop..." [16]. Looking at land from one year (such as 2007 or 2008) to another (2019 or 2020) does not capture the history of the field and will not determine how long ago this land was in a natural condition.

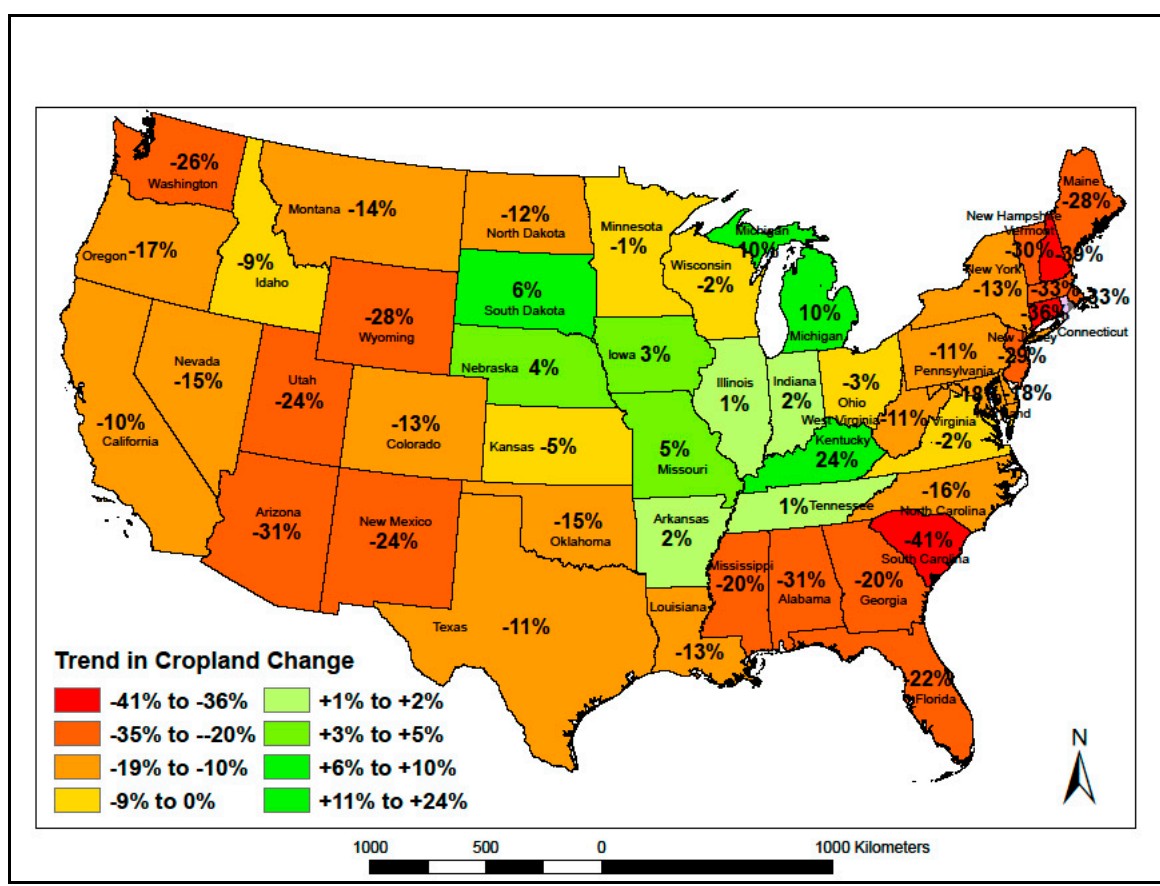

**Figure 2.** Trends in cropland from 1974 to 2012 by states, according to the USDA Major Land Uses Dataset (map created by author). Positive percentages indicate more cropland.

No USDA dataset indicates area in native lands, but USDA research has concluded native grassland would most likely be found in the rangeland category (native grasses, grass-like plants, forbs, or shrubs) of the USDA's Natural Resources Inventory (NRI) [17,18]. Increases in cropland could come from rangeland consisting of native grass species. Identifying locations with declining rangeland with a commensurate increase in cropland could lead to identifying native lands that may have been converted to cropland. However, this analysis requires further detailed analyses. Moderate resolution satellite imagery has been used in the past to try to map change from natural land to cropland in the USA.

One issue with moderate resolution satellite imagery is that it cannot differentiate native grassland from idle or former cropland or pasture [19]. For instance, the USDA's Cropland Data Layer (CDL) [20] and the National Land Cover Database (NLCD) [21], two satellite imagery-based products often used to measure land use change, do not differentiate between pasture and natural grassland. Moreover, land that has been purposely held in reserve by farmers and the US government or allowed to return to a natural state cannot be differentiated from native land using moderate resolution satellite imagery. It is also very difficult to separate grassy crops such as hay, alfalfa, and wheat from natural grasslands.

Potentially, the largest issue with moderate resolution imagery is accuracy when measuring change. Studies that have used the CDL and NLCD from one recent year to another often estimate very large changes from grassland to crop. These studies have estimated a 17% decline in grassland and wetlands in the Prairie Pothole Region of the USA from 2008 to 2017 [5], 530,000 hectares in the Western U.S. Corn Belt from 2006 to 2011 [6], and 1.2 million hectares in the entire USA from 2008 to 2012 [7]. Recent research, however, has found satellite imagery at the resolution used by these studies often overestimates change to cropland. Earlier years of the Cropland Data Layer greatly underestimate cropland, and a comparison of CDL change from year to year to year has areas of land

inconsistently move in and out of categories between the beginning and ending years chosen [8]. The CDL and NLCD also often contradict other datasets always estimating more acres of change to cropland [9]. The accuracy of the CDL has been found to be high in states with consistent crop acreage but has been found to be less than 50% in other states with more land transitions [10]. Most of these studies did not consider previous land use.

For these reasons, the use of satellite imagery at the resolution of the CDL and NLCD to estimate change to cropland should be used with caution. Both datasets consistently estimate large areas of change, and previous research efforts to correct for the errors have been shown to remove as much actual as erroneous change [8]. The Frequently Asked Questions (FAQ) provided by USDA NASS regarding the CDL specifically states "Unfortunately, the pasture and grass-related land cover categories have traditionally had very low classification accuracy in the CDL . . . We recommend users consider using the USGS NLCD (https://www.mrlc.gov/; accessed on 28 April 2022) for research involving non-agricultural categories and grassland/pasture categories." [22].

### 1.2. Present Study

As noted, previous satellite research has focused on static landcover change typically from 2007 or 2008 to present day. This does not enable a long-term analysis of land cover and use. The studies often provide estimated carbon penalties from the conversion of natural lands to crop, which will be much higher than idle cropland returning to production. Given the complexity of identifying land use change using any one dataset, especially moderate range satellite imagery, this study used historical data across distinct data sources, analyzing multiple USDA tabular datasets to identify locations where rangeland is declining and cropland increasing. Recognizing that each dataset was collected from different sources and may not be temporally or categorically equivalent, the study only used the tabular data to find likely locations of change not to estimate change. Once these potential land use change areas were identified, moderate resolution satellite datasets and USDA aerial imagery were used to find areas of change from natural lands to crop versus areas reflecting former crop that has been fallow or idle. In addition, the study analyzed spectral curves of vegetation indices for these areas from 1985 to 2020 to help understand historical land cover. It is hoped this study will provide a more accurate way to quantify land cover change to and from crop by using the combined tabular datasets, historical moderate resolution satellites, and aerial imagery.

## 2. Materials and Methods

### 2.1. Materials

Unless otherwise stated, all data used are available on public websites hosted by the USDA, except for the NLCD data, which are also available publicly but are developed from a consortium of government agencies led by the United States Geological Survey (USGS). Analysis of the tabular data was performed at the U.S. County level (Table 1).

**Table 1.** USDA datasets used in the study with the agency, their publication schedule, and primary use within the USDA. * The USDA Farm Services Data was available to the public at the county level since 2008. The authors requested and received 2007 from the USDA Farm Services Agency.

| Dataset | Agency | Schedule/Availability | Primary Purpose |
|---|---|---|---|
| USDA Farm Services Agency Crop Totals | Farm Services Agency | Annual since 2007 * | Track producer acreage for USDA program compliance |
| USDA Natural Resources Inventory | Natural Resources Conservation Service | Every 5 years since 1982 | Track land cover/use and conservation practices in the USA |
| USDA Census of Agriculture | National Agricultural Statistics Service | Every 5 years since 1840 | Track farm/crop land, uses, and farm techniques |

**Table 1.** *Cont.*

| Dataset | Agency | Schedule/Availability | Primary Purpose |
| --- | --- | --- | --- |
| USDA Cropland Data Layer | National Agricultural Statistics Service | Annual for all of USA since 2008 | Calculate crop acreage and assist in sampling |
| National Land Cover Database | Multi-Resolution Land Characteristics Consortium | 2001, 2004, 2006, 2008, 2011, 2013, 2016, 2019 | Provide nationwide land cover and change for the USA |
| USDA National Aerial Imagery Program | Farm Services Agency | Typically biennial by state since 2003 | Used by FSA to confirm program compliance |

2.1.1. Dataset 1: USDA Farm Services Agency (USDA FSA) County-Level Crop Acreage Data

Each year, agricultural producers in the USA who participate in any USDA program must report the crop and acreage for the planted fields to the USDA Farm Services Agency (FSA). This includes federal crop insurance programs and loan programs. A very high percentage of total agricultural land is reported. From 2008 to 2020, the FSA national soybean planted acreage was always lower than the USDA National Agricultural Statistics Service (NASS) statistical reporting data but was always within 2% (Figure 3) [14,23]. NASS uses the FSA data to assist in the compilation of these reports. The acreage is compiled and reported by the FSA at the county, state, and national level and made available to the public. County-level data have been publicly available since 2009. For this study, a request was made and granted by FSA for the 2008 county-level data.

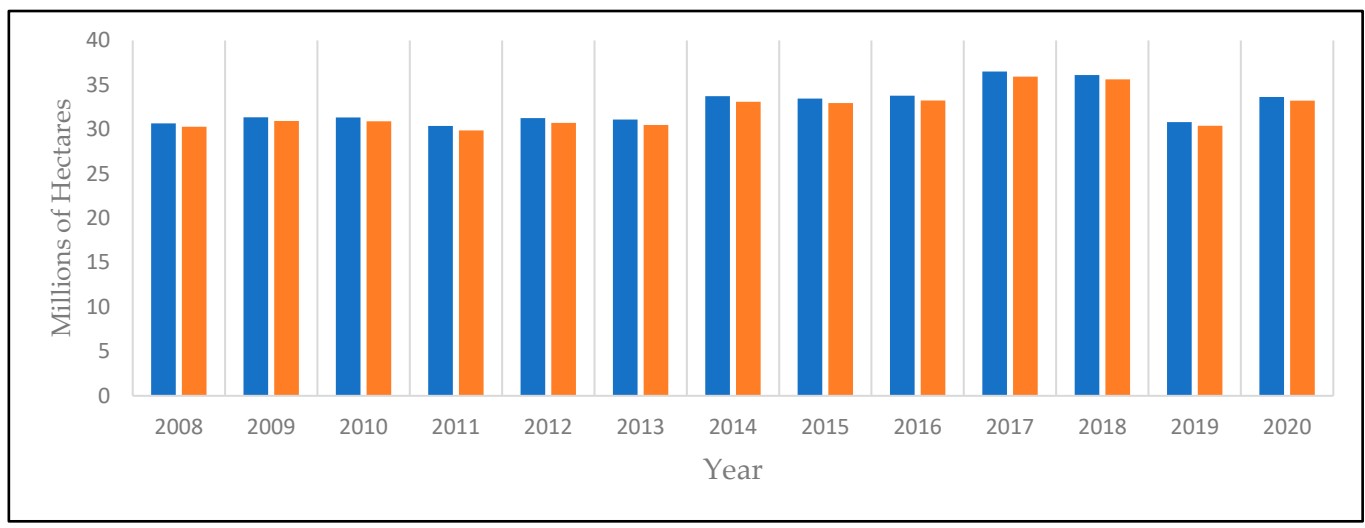

**Figure 3.** Total Soybean Hectares planted in the USA from USDA NASS and USDA FSA.

Because the data are compiled directly from a very large percentage of growers, the FSA crop area data are, most likely, the most complete, granular dataset for crop planting area in the USA. Comparison of changes in crop area in a county from year to year may indicate potential conversion of native lands. This study compared crop area by county in 2008 to crop area in 2020. The starting year of 2008 was chosen, as this was the year many regulations went into place, and it was the first year the CDL and FSA county-level data were available for the entire continental USA. FSA data also include the CRP area. CRP hectares are cropland set aside by growers with financial support from the USDA to provide areas to reduce runoff and promote natural vegetation and wildlife. From 2008 to 2020, the funding available for CRP declined, reducing enrollment. Many of these hectares likely returned to agriculture. The FSA data should help in understanding conversion of CRP to active cropland.

2.1.2. Dataset 2: USDA Census of Agriculture Data (Census) (USDA NASS, 2017)

The USDA Census is conducted every five years [24]. It has been compiled since 1840. Information from a large proportion of agricultural producers is collected from question-naires and through statistical techniques, estimating data across the entire agricultural population. Official Census reports contain the results at the county, state, and national level. These data include total cropland, harvested cropland, cropland/pasture, idle crop-land, summer fallow cropland, and many other land cover categories. This study examined the idle cropland and cropland/pasture categories for 2007 and 2017. The limitations of the dataset were recognized, including the error margins and temporal differences when comparing with the FSA data (2007 and 2017 compared to 2008 to 2020); however, the study felt it was appropriate to use the data, since the results were not used for direct estimations, only as a guide to better understand the possibility of conversion of natural grasslands to cropland.

The Census defines cropland/pasture as "pasture and grazing land that could have been used for crops without additional improvements". This land, when combined with idle cropland could represent readily available lands for crop. It might make more sense for a producer to use this land prior to converting natural lands. The study determined changes from 2007 idle cropland and cropland/pasture total hectares from 2017 idle cropland and cropland/pasture at the county level. These were then compared to total hectares increasing in cropland for the remaining counties from the FSA data. If there was a greater decline in land from these two categories than the increase in cropland, the counties were removed from consideration for potential conversion of natural land as, most likely, any new cropland came from idle or cropland/pasture.

2.1.3. Dataset 3: USDA Natural Resources Inventory (NRI) (USDA NRCS, 2017)

Rangeland area at the county level was analyzed from the NRI. The NRI has been developed every five years since 1982 by the USDA's Natural Resources Conservation Service. Estimates for different land covers within the USA are calculated based on the results obtained from the analysis of high-resolution aerial photography (15 cm) and ground information for over 800,000 points. Data are released at the state and national level. County-level data are available upon request, but the USDA warns that error margins at the county level may be greater than the actual values for different land covers. Recognizing this caution and the limitations of the dataset due to the error margins and temporal differences when comparing to the FSA data (2007 and 2017 compared to 2008 to 2020), this study used NRI data at the county level for 2007 and 2017, focusing on rangeland area. Since it is recognized by the USDA as the best dataset for rangeland analysis, use of NRI data was considered appropriate to use as a guide to counties with potential declining rangeland and not as a direct measurement of rangeland change. That is, rangeland total hectares from 2007 were compared to rangeland total hectares for 2017 for counties, which showed an increase in crop hectares from 2008 to 2020, according to the FSA county-level crop data. Counties, which indicated a decline in rangeland and an increase in cropland, were included in further analysis.

2.1.4. Dataset 4: USDA Cropland Data Layer (CDL)

The CDL has included satellite images to interpret land cover across the continental USA (48 states) annually since 2008. Several agricultural states also have CDLs from previous years. The images are classified into 134 land cover or use classes. The USDA only classifies crops and uses the NLCD for other land categories. Accuracies for identifying major crops by the CDL have been reported as high as 90% or above, but as low as 50% for identifying noncropland classes [25]. Errors associated with the noncropland land cover indicates against the use of the CDL as a standalone tool for land cover change to cropland. This study only used the CDL to identify areas of potential change for further investigation.

2.1.5. Dataset 5: MLRC National Land Cover Database (NLCD)

The NLCD is a satellite-based classification of land cover for the continental USA. It has been developed every five years, but data for intermittent years have also been released (2001, 2006, 2008, 2011, 2013, 2016, and 2019). The NLCD is developed by a consortium of government agencies, which contribute ground information on land cover and change. It is developed using Landsat 30-m resolution imagery. The USDA recommends the use of the NLCD, as opposed to the CDL, in non-agricultural areas, as these areas are derived from the NLCD within the CDL. Similar to the CDL, the NLCD tends to overestimate change and was only used as a guide for locations for further investigation in this study.

2.1.6. USDA National Aerial Imagery Program (NAIP)

This USDA program offers aerial imagery collected in agricultural states on an annual or biennial basis. The spatial resolution of the imagery ranges from 1 to 2 m, in either visible bands (red, green, and blue) or visible and near-infrared bands (red, green, blue, and near-infrared) [26]. For this study, a visual assessment of NAIP from 2007 to 2020 (depending on availability for each state) was used to attempt to identify conversion from other land cover to crop and conversion from crop to other land cover.

*2.2. Methods*

2.2.1. Identify Locations with Increasing Cropland

A comparison was made between total area in cropland for 2008 and 2020 using the FSA's county-level crop data. The FSA estimates are broken down by specific crops including CRP acreage, allowing for a detailed assessment of crop acreage at the county level. For this study, every county in the USA was analyzed to determine if crop acreage had increased and which crops were increasing from 2008 to 2020. Hectares in CRP and fallow land were considered a source of available agricultural land and were removed from any cropland increase.

2.2.2. Identify Available Lands

Lands were identified that were readily available for conversion to crop for the counties with the increasing FSA cropland from 2008 to 2020. Readily available lands included county-level cropland/pasture and idle lands from the 2007 and 2017 Census. Cropland/pasture in the Census is land considered in pasture that can be converted to crop without additional improvements. There is another category of pasture that includes land in permanent pasture, but this was not considered readily available for conversion to crop in the study, since it might require as much work as converting natural grassland. If there were declines in idle or cropland/pasture, these were subtracted from increases in crop (i.e., these would be the easiest lands to convert back to agriculture).

2.2.3. Identify Counties with Increasing Cropland and Declining Rangeland

The NRI was used to determine if the remaining counties had any rangeland and, if yes, whether there was a decline in rangeland from 2007 to 2017. This decline in rangeland was then compared to the change in cropland. If a county had an increase in cropland from 2008 to 2020 and also had a decline in rangeland, it was considered a candidate for conversion from natural land to cropland and analyzed further.

2.2.4. Analyze NAIP Imagery for Land Cover and Change

NAIP imagery from 2007 to 2020 was visually compared in Google Earth Engine on a parcel-by-parcel basis in each of the remaining 22 counties to determine whether land potentially with native species (no signs of cultivation) was converted to crop. Previous studies have found manual interpretation, especially in complex landscapes where a lot of change occurs to be more accurate than automated methods [27]. NAIP is typically collected biennially, and some of the states with counties of interest did not have imagery in

2008 or 2020 (Table 2). Scripts were written in Google Earth Engine which allowed multiple NAIP years to be overlaid, with the ability to toggle back and forth between years.

**Table 2.** Years of NAIP Availability for Each State with Potential Cropland Conversion Counties.

| State | Years of NAIP Availability |
|---|---|
| Kansas | 2008, 2010, 2012, 2014, 2015, 2017, 2019 |
| Nebraska | 2007, 2009, 2010, 2012, 2014, 2016, 2018, 2020 |
| Oklahoma | 2008, 2010, 2013, 2015, 2017, 2019 |
| South Dakota | 2008, 2010, 2012, 2014, 2016, 2018, 2020 |
| Texas | 2008, 2009, 2010, 2012, 2014, 2016, 2018, 2020 |

Estimated change to cropland from 2008 to 2020 in the CDL and NLCD was used as a guide for potential areas of change. The CDL was generalized into crop and non-crop classes. Any pixel that was identified as non-crop in 2008 and crop in 2020 was highlighted in red in Google Earth Engine (i.e., change to crop). In contrast, lands that were in crop in the 2001, 2004, and 2006 NLCD but grassland in the CDL in 2019 and 2020 were flagged and colored green in the overlay of the NAIP images in Google Earth Engine (i.e., conversion from crop).

An additional layer used in the analysis was the Bureau of Land Management's Public Land Survey System (PLSS), which is a uniformed size grid overlay for the USA [28]. With the PLSS, any land in the USA can be identified with a specific letter/number combination. For the study, the viewer extent was set to a Section level, which is one square mile (259 hectares/640 acres), and that land was examined. Beneath the CDL/NLCD change layers and PLSS were the 2008 to 2020 NAIP images (Figure 4). For each county, the analysis would examine the map section, look for change identified by the CDL/NLCD, and then observe each year of the NAIP for land cover change. If there were no signs of current or past cultivation in the 2008 NAIP, but later years showed cultivation, a polygon was drawn around the area of change and total hectares of potential change were calculated. Likewise, if the NLCD/CDL layer showed potential change to grass, the area was reviewed in the NAIP images. To ensure temporarily fallow fields were not included, it was required that the field show continuing vegetation growth (no crop) for six years in the NAIP imagery. The CDL/NLCD tended to show extensive conversion where the NAIP did not, but larger, field size areas of change in the CDL/NLCD often were also seen as change in the NAIP imagery.

To ensure natural areas were also identified, the USGS Protected Areas Database (PAD-US) GIS layer was placed over the NAIP images [29]. The PAD-US contains the locations of many different types of protected lands. These lands should be in a natural state in all NAIP years. Time was spent observing the appearance of these locations on the NAIP imagery prior to analysis. Typically, natural areas have a rougher texture and lack the uniform growth seen in cultivated cropland. Still, it was recognized that hay, CRP, and some cereal crops could appear as natural lands. A parcel was only considered converted to crop if it had no signs of previous cultivation (signs of rows, terraces, uniform crop growth patterns) in 2007 or 2008 and was cropland in 2019 or 2020. Figure 5 lists the steps of the analysis with short descriptions.

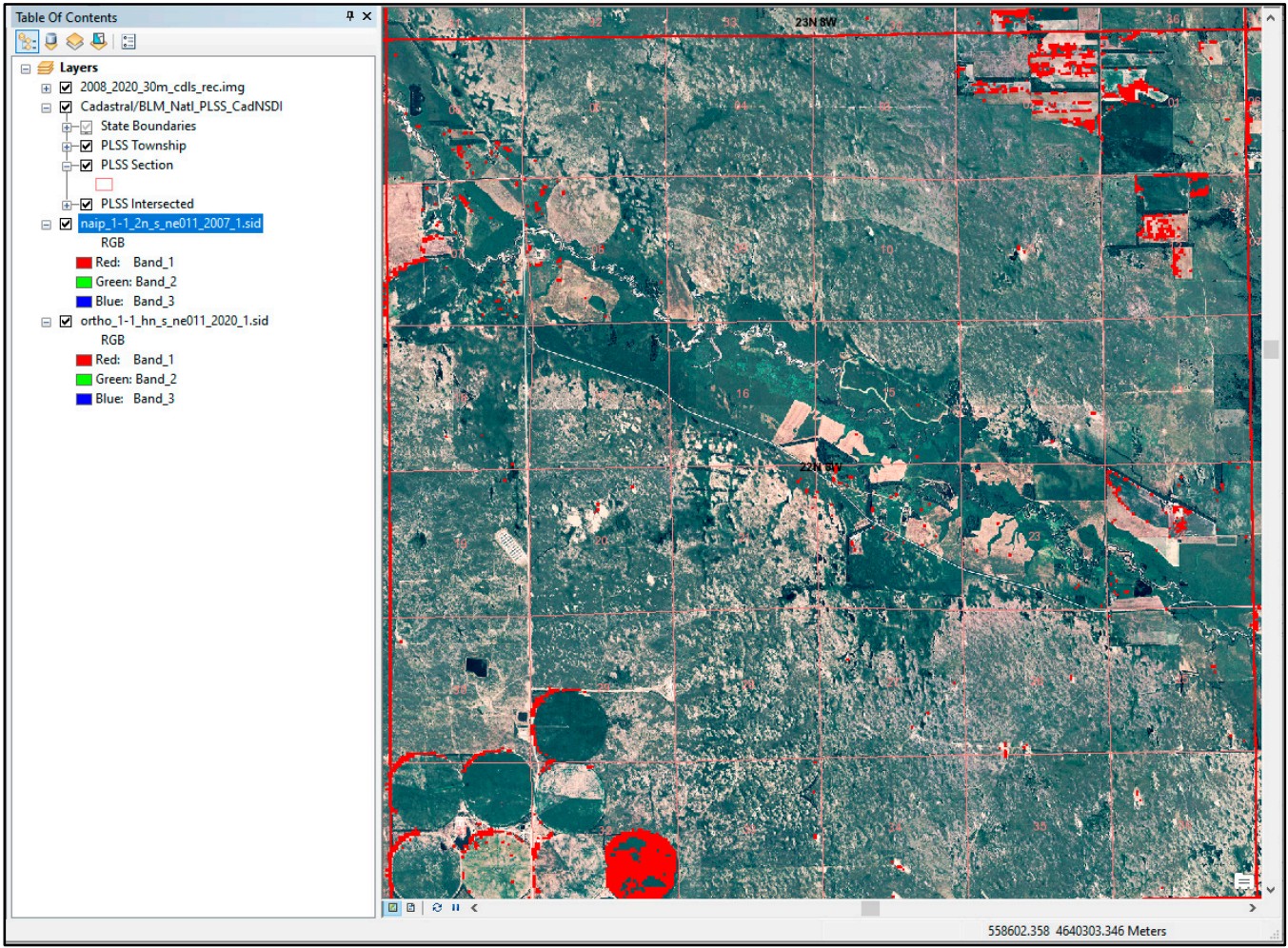

**Figure 4.** Example of viewer settings and layers for a county in Nebraska.

2.2.5. Analyze Change Parcels Using LandTrendr to Understand Historical Land Use

If land met all the previous criteria, the parcel was examined in Google Earth Engine using LandTrendr's Zonal Time Series Plotter. LandTrendr was developed at Oregon State University and uses temporal segmentation of Landsat 30-m satellite imagery to identify areas of land cover change [30]. For this study, Zonal Time Series Plotter was modified to generate three normalized difference vegetation index values per year (compared to one in the default). The following indices were generated: spring index (20 March to 30 May), summer index (10 July to 30 August), and fall index (30 September to 30 November) for each year from 1985 to 2021. The study also modified the Time Series Plotter to plot the mean values, instead of a single pixel, for the fields identified as change.

If the CDL/NLCD analysis, NAIP imagery, and LandTrendr analyses all indicated grassland from 1985 to 2007/2008 and conversion to crop in a later year, a polygon was drawn in Google Earth Engine around the boundary of the field to calculate hectares of change to crop. Conversely, if land was identified as crop between 1985 and 2007/2008 and was identified as grass/fallow in at least three consecutive NAIP image years (because NAIP is biennial for a total of 6 years), a polygon was drawn around this field to calculate hectares change to grass.

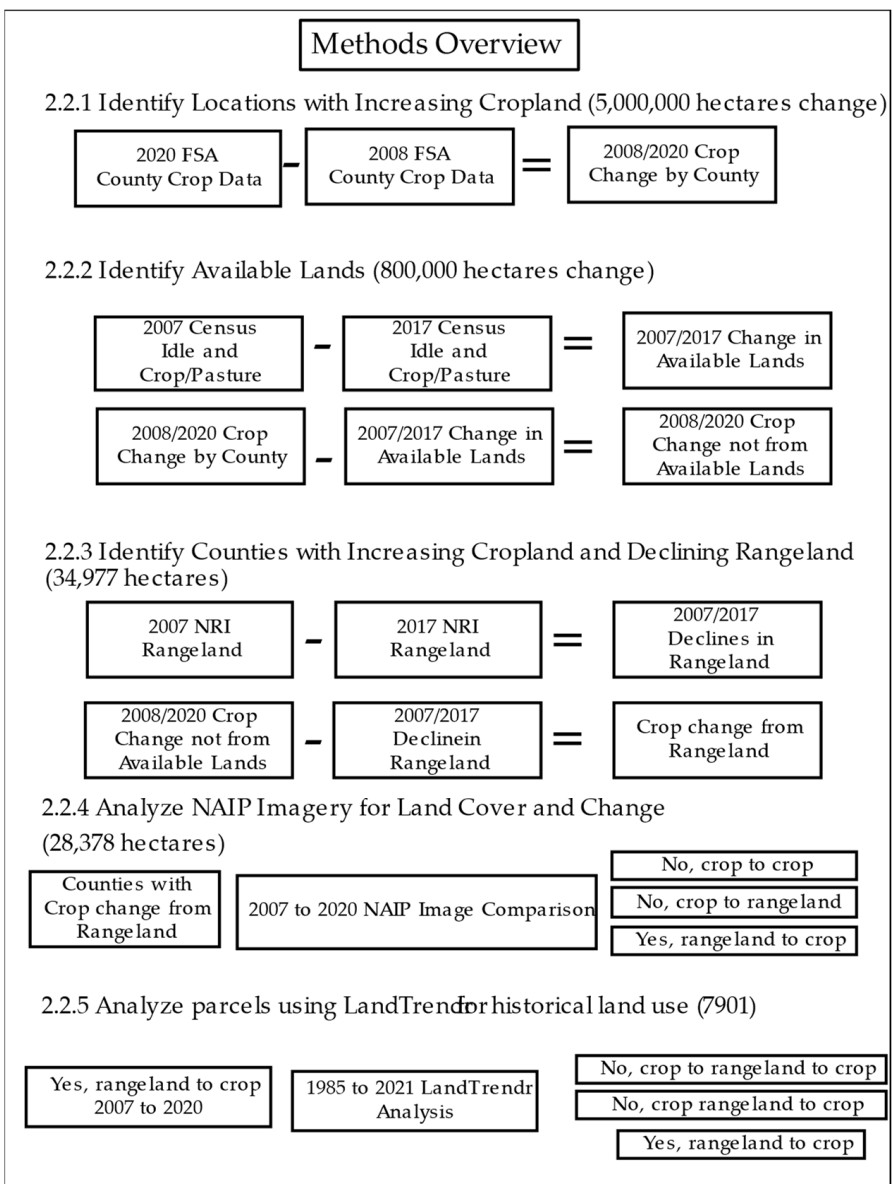

**Figure 5.** Steps and methods for each part of the analysis.

## 3. Results

### 3.1. Identify Locations with Increasing Cropland

According to the FSA dataset, overall hectares for major crops declined from 2008 to 2020 in the USA by almost 1.5 million hectares (Table 3). However, specific crops, such as soybeans and corn, increased by a combined five million hectares. These acres could have easily come from acres formerly used for wheat, which declined by over six million hectares, or from CRP, which declined by over five million hectares.

**Table 3.** USDA Farm Services Agency Total Hectares in Major Crops in 2008 and 2020 (major crops include barley, corn, cotton, oats, rice, sorghum, soybeans, sugar beets, sugarcane, and wheat).

| USDA FSA Data | 2008 Hectares | 2020 Hectares | Difference |
|---|---|---|---|
| Major Crops | 100,407,942 | 98,920,061 | −1,487,880 |
| Soybeans/Corn | 64,019,254 | 69,092,758 | 5,073,504 |
| Wheat | 24,916,943 | 18,716,925 | −6,200,018 |
| CRP | 14,092,072 | 8,722,711 | −5,369,360 |

Thus, a county-by-county examination provided details on specific counties where corn, soybeans, or other crops increased in acreage, beyond any decline in other crops or declines in CRP, which would require land from another source (idle cropland, pasture, hay, etc.).

In total, 919 counties saw an increase in cropland. This totaled over five million hectares (Figure 6). Over 2000 counties, however, saw declining crop acreage totaling just over 16 million hectares. Movement was seen throughout the country, with pockets of consistently increasing crop hectares in Missouri, southeastern Iowa, central North Dakota, and central Kentucky.

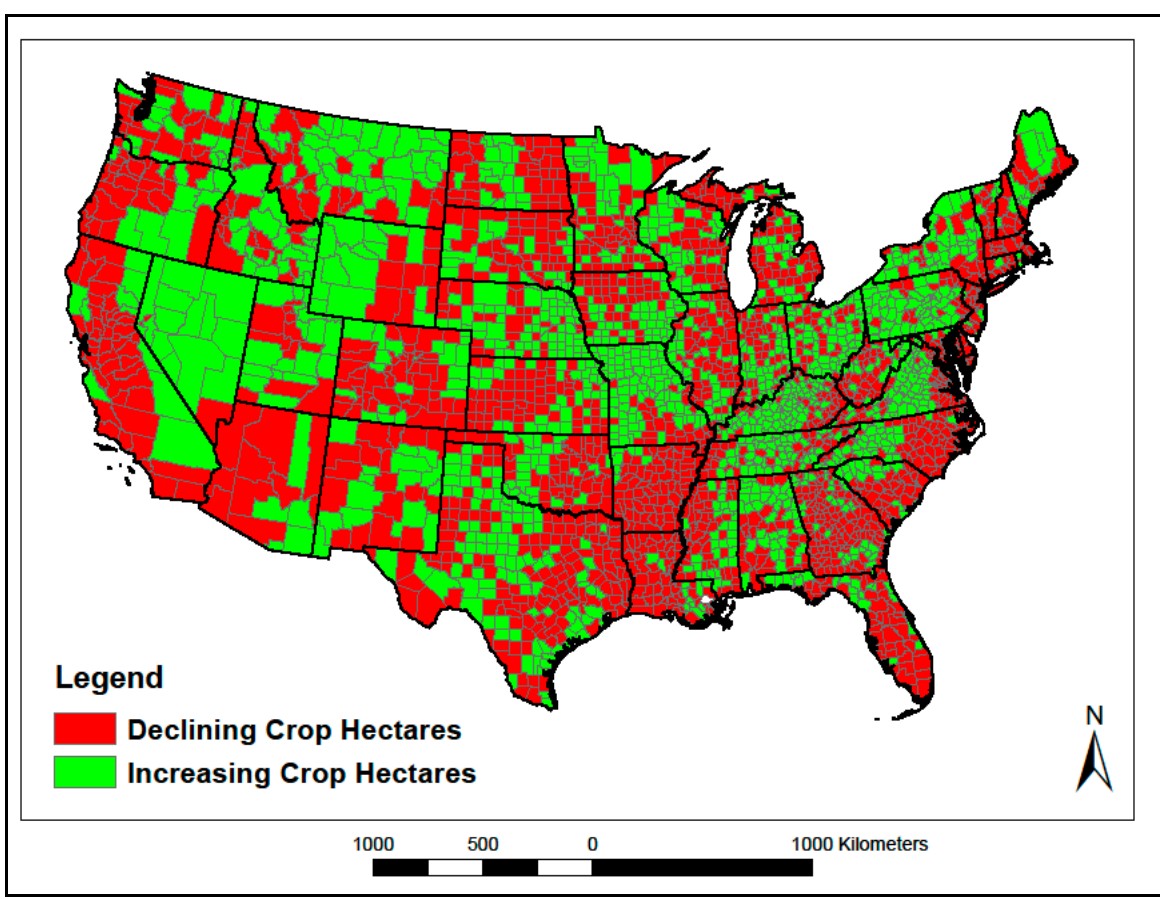

**Figure 6.** Counties with Increasing and Declining Crop Area from 2008 to 2020. Map by author.

*3.2. Identify Available Lands*

For the next step, available lands that were already appropriate for crops (idle, cropland/pasture) in the Census were calculated for each county that showed increasing cropland in the FSA data. These were counties where idle cropland or cropland pasture declined from 2007 to 2017. This land would be a likely source of previous cropland available for conversion.

Removing the readily available areas led to a decline in total hectares of change estimated, from over 5 million to just over 800,000; and the number of counties estimated declined to 249 from 919 (Figure 7). Counties were spread throughout the midwestern USA, with the greatest increase in cropland, after considering available lands, occurring in Nebraska, Kansas and Oklahoma.

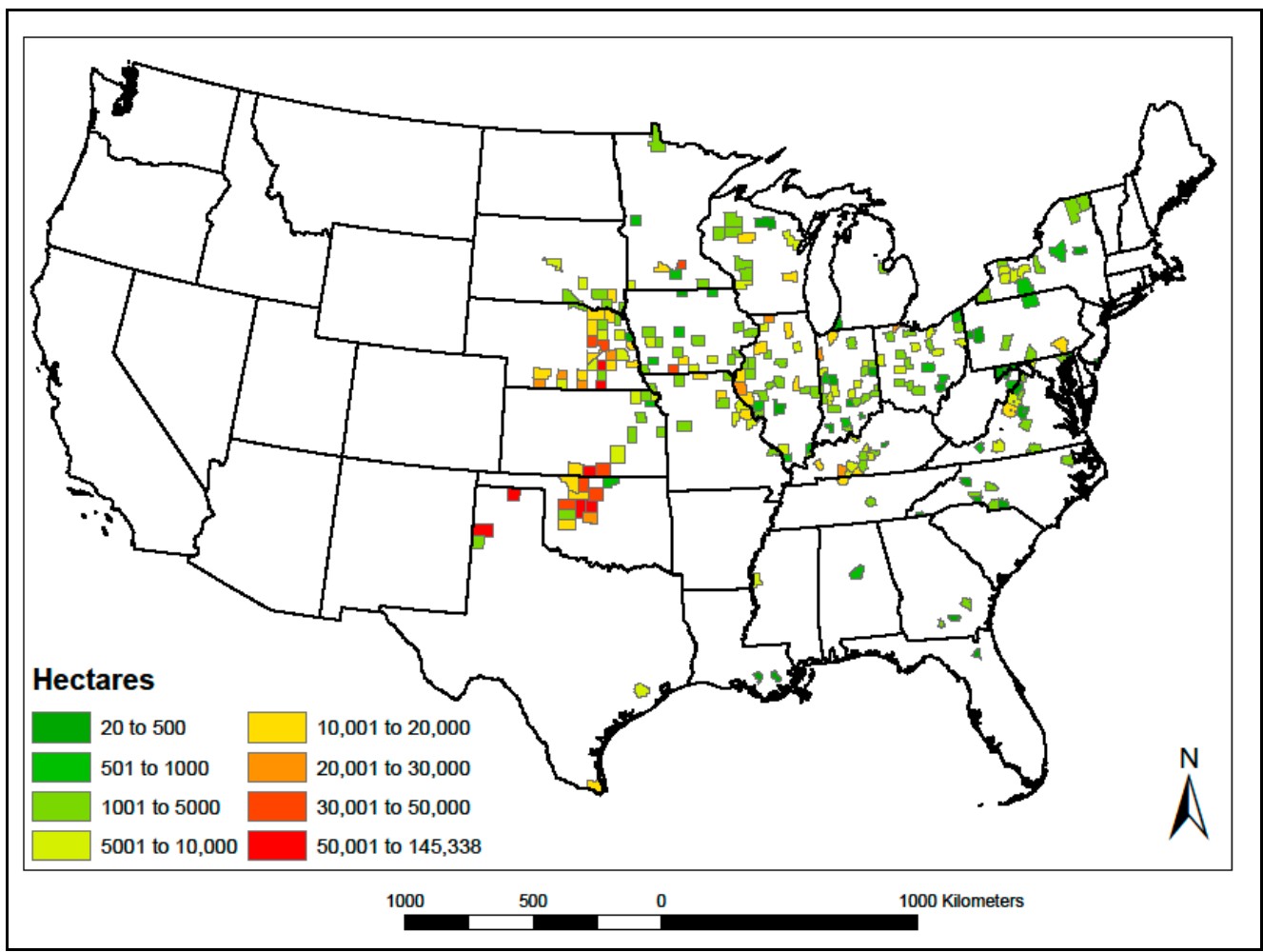

**Figure 7.** Counties with Increasing Cropland after Removing Available Lands. Map by author.

### 3.3. Identify Counties with Increasing Cropland and Declining Rangeland

As stated, analyses by the USDA have concluded that any native grassland would most likely be in the rangeland category in the NRI. Most natural land in the eastern USA was originally forest, not rangeland. Most grassland was found in what was originally the Great Plains, west of the Mississippi. Figure 8 shows the outlines of counties with existing rangeland, according to the 2017 NRI. Figure 8 also shows counties remaining in the study with both increasing cropland and declining rangeland. Most of the counties with increasing cropland were in the eastern and midwestern USA, where there is no rangeland. After the rangeland analysis was performed, 22 counties with 34,977 hectares in increasing cropland remained (Figure 9).

### 3.4. Analyze NAIP Imagery for Land Cover and Change

As discussed in the methods section, NAIP imagery along with NLCD/CDL change masks were loaded into Google Earth Engine. Land for all 22 remaining counties with increasing cropland plus 10 additional counties identified as having the greatest decline in cropland were visually assessed for change. Total hectares of change for each category were calculated resulting in 28,378 hectares of change to crop. This number was separate from the analysis of the tabular datasets and was only based the NAIP analysis. It is promising that it was close to the value from the tabular data (34,977 hectares).

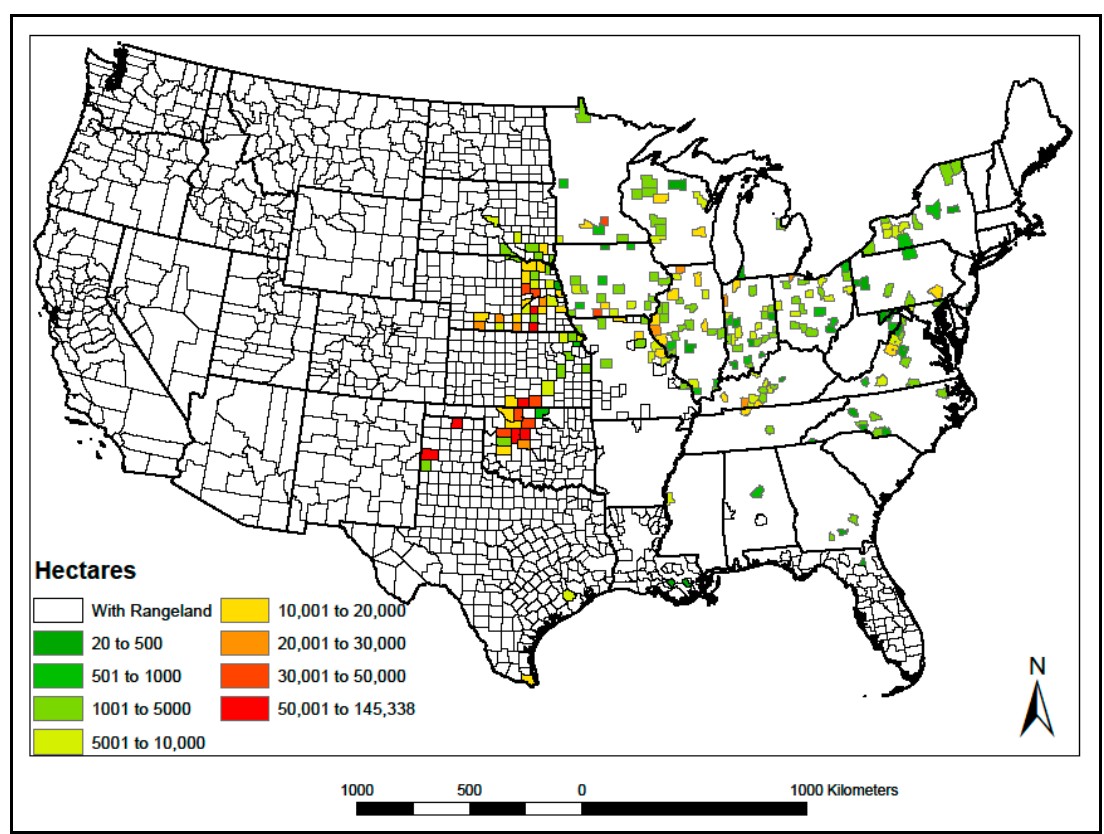

**Figure 8.** Counties with Rangeland and Increasing Cropland. Map by author.

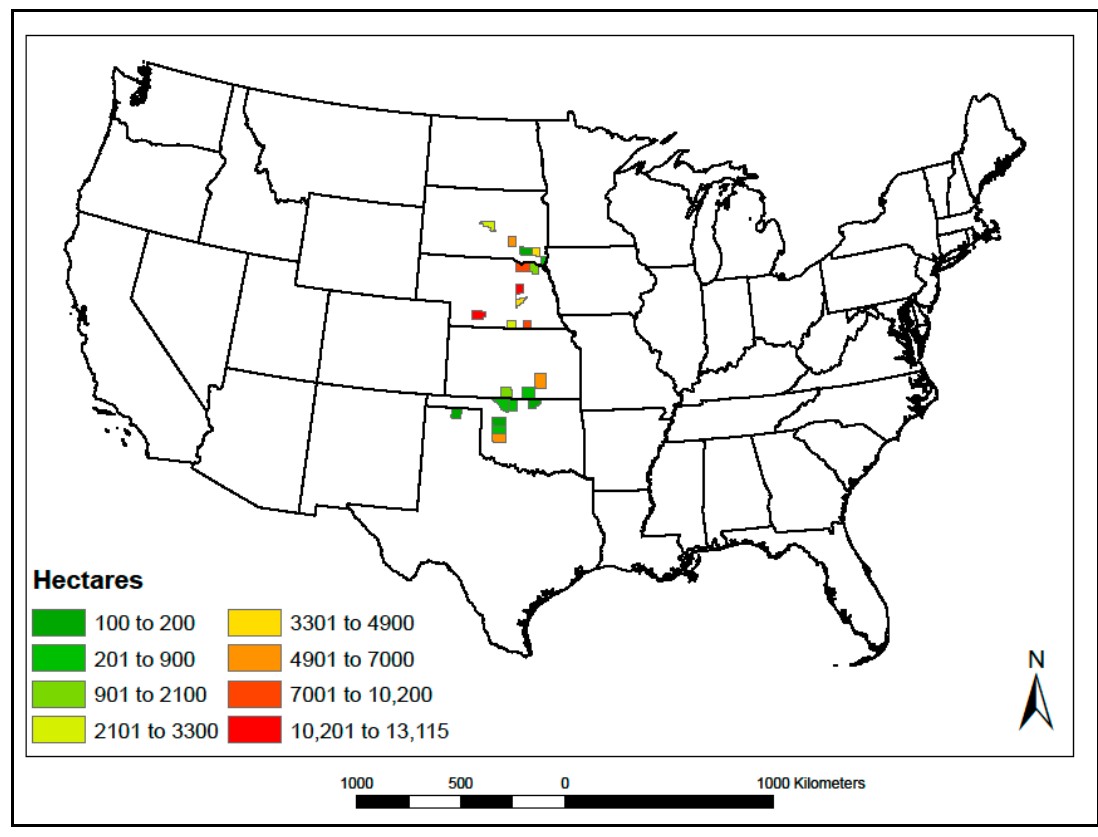

**Figure 9.** Counties with Increasing Cropland and Declining Rangeland or Forest. Map by author.

The temporal curves from the LandTrendr algorithm often identified land transitions from crop to fallow to grass and vice versa during the 1985 to 2021 time period (Figure 10). Crop has a distinctive annual signature of low values in the spring and fall and high values in the summer, while grass will have a flatter curve (more growth in spring and fall but less in the summer). Crop will be bare soil or residue in the spring and fall and have maximum growth in the summer. Grass will be present in the spring and fall giving some value to the vegetation index but will not be as vibrant as the crop in the summer. LandTrendr analysis allowed estimation of whether land in grassland in 2007/2008 was previously in crop back to 1985. The NAIP images from 2008 to 2020 were used as training guides in identifying the field uses from 1985 to 2007, and NAIP and other aerial image sources such as the National High Altitude Photography [31] and the National Aerial Photography Program [32], which have imagery from the early 1980s to late 1990s, were used to verify some of the curves back to 1985.

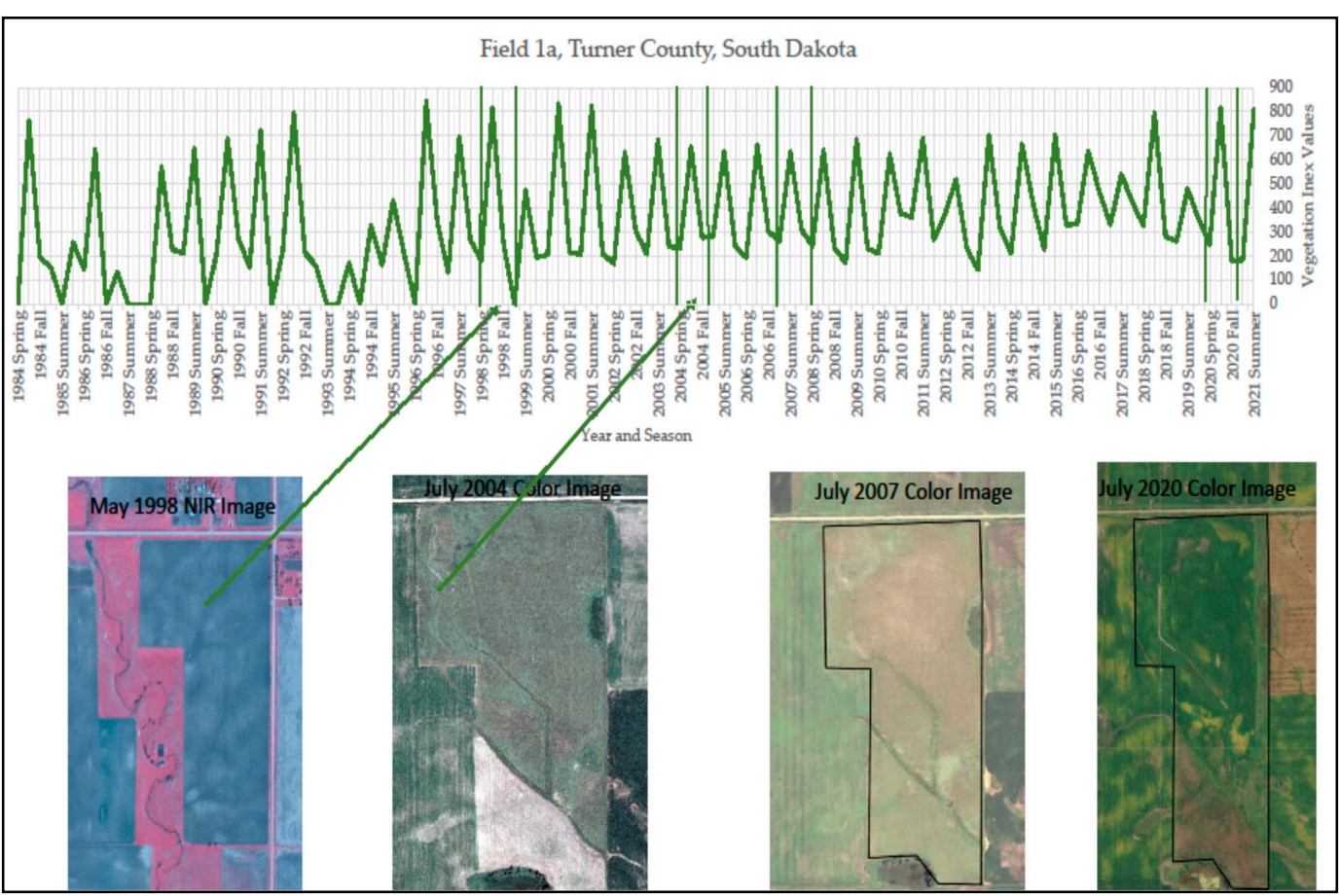

**Figure 10.** Example of LandTrendr temporal vegetation index signature that was crop until early 2000s, became fallow and returned to crop in 2014.

The 22 counties identified with increasing crop acreage had similar hectares converted to crop as those reverting to grassland from 1985 to 2020 (Table 4). Additionally, the selected ten counties with declining cropland had many more acres reverting to grassland.

**Table 4.** Total Hectares found to have potentially changed to and from cropland in the 22 studied counties since 1985.

| Name | Final Change to Crop (Hectare) | Final Change to Grass (Hectare) |
|---|---|---|
| Alfalfa, Oklahoma | 166 | 42 |
| Aurora, SD | 198 | 192 |
| Barber, Kansas | 294 | 598 |
| Boone, Nebraska | 1011 | 118 |
| Butler, Kansas | 232 | 803 |
| Cedar, Nebraska | 205 | 78 |
| Custer, Oklahoma | 12 | 50 |
| Dewey, Oklahoma | 150 | 358 |
| Frontier, Nebraska | 76 | 51 |
| Hansford, Texas | 115 | 329 |
| Hughes, SD | 245 | 312 |
| Hutchinson, SD | 176 | 737 |
| Kay, Oklahoma | 59 | 359 |
| Knox, Nebraska | 3219 | 361 |
| Merrick, Nebraska | 74 | 206 |
| Sumner, Kansas | 300 | 0 |
| Thayer, Nebraska | 365 | 35 |
| Turner, SD | 251 | 275 |
| Union, SD | 295 | 50 |
| Washita, Oklahoma | 6 | 820 |
| Webster, Nebraska | 351 | 246 |
| Woods, Oklahoma | 100 | 127 |
| Total | 7901 | 6145 |

## 4. Discussion

Previous studies have used combinations of moderate resolution satellite imagery, USDA tabular datasets, and high-resolution USDA aerial photography to attempt to quantify changes from natural land to cropland in the USA. However, each of these datasets has limitations. Moderate resolution satellite imagery is hindered by low accuracies for noncropland classes and limited land cover/use classes. Previous studies have also not used historical imagery to track previous land use, assuming the land was in its current cover/use for a long period. Tabular datasets are hindered by granularity and limited land cover/use classes. High resolution airborne imagery is effective at interpreting land cover with excellent granularity and allows for a greater breakout of land cover/use classes; however, analyzing the entire USA would be very time and labor intensive.

This study used the strengths of each type of dataset, using the tabular data to find counties with potential change to cropland, moderate resolution imagery to find potential change locations at the field level, and historical land cover/use and high-resolution airborne imagery to determine change at or below the field level. By using tabular USDA datasets on crop and available land, the study was able to focus more detailed analysis on specific locations with the greatest potential for conversion of natural land to cropland. The satellite-derived land cover classifications helped identify these potential locations but also provided many false positives, which, if used alone, would have led to inflated rates of conversion to crop. Cropland change, however, is not static. Land going in and out of

crop can vary from year to year. Adding the LandTrendr temporal vegetation index curves allowed the study to review land use from 1985 onward. Temporal curves for known land use from observation of NAIP images (crop, grass, forest, fallow, hay) in more recent years could be applied to identify land use in earlier years where NAIP was not available. This led to the discovery that, most likely, much of the land converted to crop from 2007/2008 to 2019/2020 was previously in crop. Of the close to 35,000 hectares identified as changed prior to this analysis, only 7901 hectares did not show evidence of previous crops.

This could indicate a potential balance across time in the overall crop footprint in the USA, with marginally productive lands going in and out of row crop production, hay, fallow, and idle based on several factors including pricing. A look at harvested hay hectares in the twenty-two counties the study analyzed supports this, with hay reaching the highest area in the counties in 2001 with a slight rise in 2007 and steady decline thereafter (Figure 11) [33]. A decline in area planted to hay seems to coincide with the increase in crop hectares seen in 2007/2008.

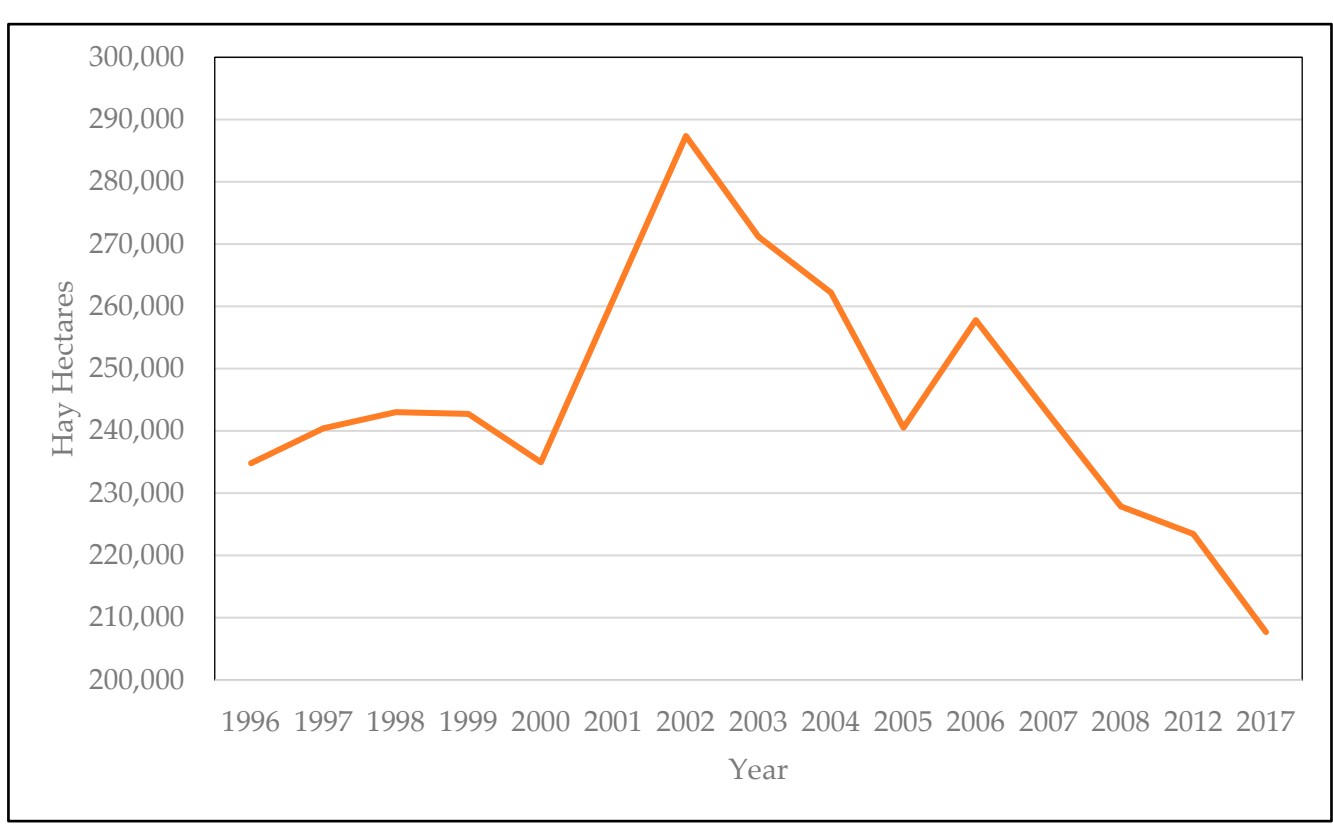

**Figure 11.** Harvested Hay Hectares for the 22 counties in the study since 1996.

It is interesting to note that a similar indicator was seen in the temporal field signatures from LandTrendr (Figure 12). Many fields identified as change from grassland to cropland from 2008 to 2020 showed earlier signs of being in crop before going dormant/to grass starting in the late 1990s or early 2000s.

This also coincided with the corn and soybean net values calculated by the USDA (Figure 13). The net value of both corn and soybeans declined into the negative in the late 1990s [34]. This most likely triggered a reduction in acres planted as a response to the crops losing profitability.

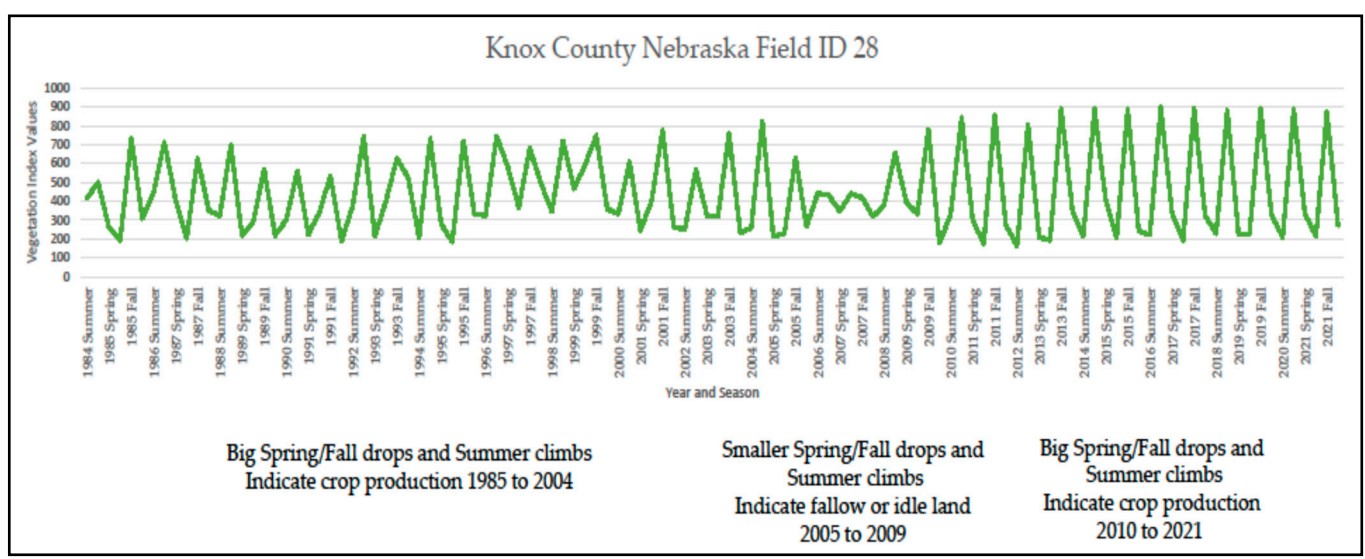

**Figure 12.** LandTrendr temporal signature for a field that was identified as change from grass to crop between 2008 and 2020 but showed signs of cropland in earlier years.

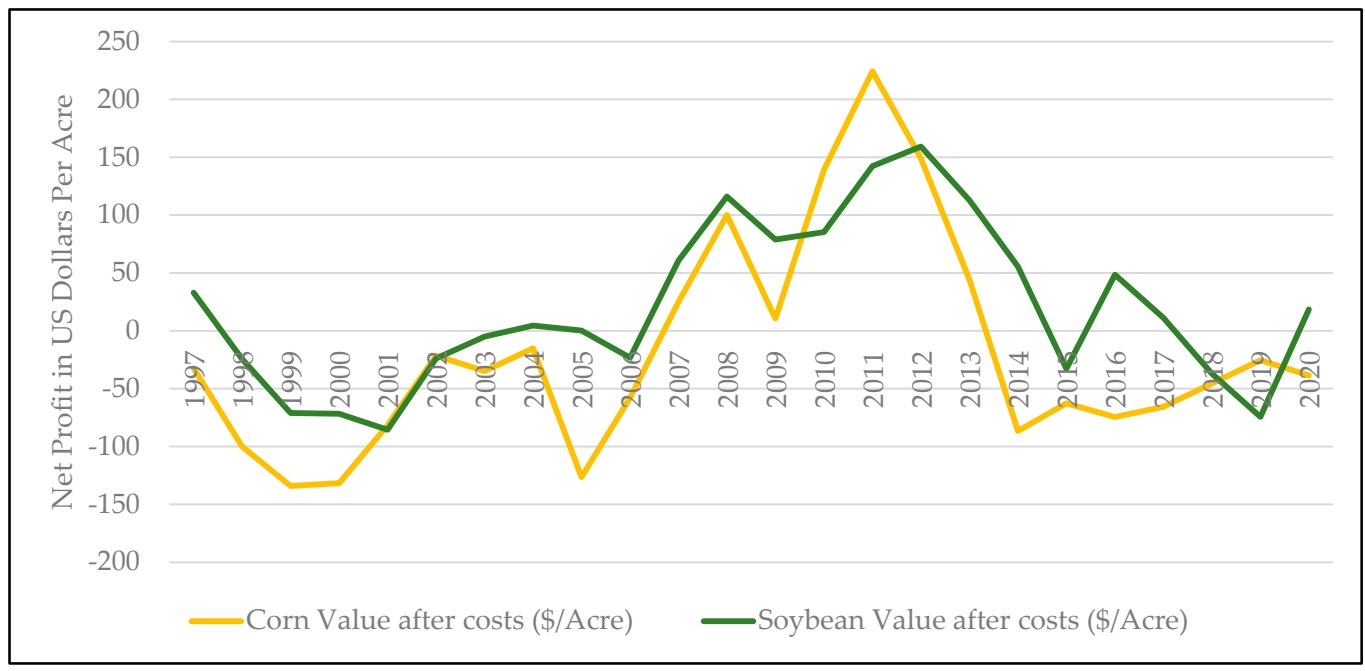

**Figure 13.** Net value of soybean and corn ($/Acre).

Then, the net values of the crops started to rise in 2006, just before many studies that tried to estimate land cover change were initiated. Land observed in satellite imagery in 2007 or 2008 that was left idle from the early 2000s may appear to be native in moderate resolution satellite imagery from 2007 or 2008. Additionally, demand models have indicated that yield improvements have also offset the need for additional acres, which may explain why the net profit curves return to negative in 2018/2019, as too much crop was being produced [35].

**5. Conclusions**

Government agencies in the USA and the European Union have run their own analyses with data from studies using moderate resolution satellites, which indicate large amounts of land has moved from native land to agriculture as a result of the introduction of biofuels.

This land conversion has a steep carbon impact, and these agencies will not subsidize these fuels or grains if it is shown that large areas of native lands were converted. Correct identification of natural lands is critical to their protection. Overestimation not only unnecessarily penalizes biofuels and agriculture but can make it difficult to identify land that needs to be preserved.

Using USDA tabular datasets, satellite data, and aerial imagery to attempt to locate United States counties with a high likelihood of conversion to crop allowed for detailed analyses of individual land parcels. The USDA FSA crop dataset was used to determine counties where crop increased from 2008 to 2020. Census data identified counties where cropland in pasture or idle cropland might be available for crops. USDA NRI was used to identify counties where rangeland declined from 2008 to 2020 and cropland increased. This led to the identification of 22 counties in South Dakota, Nebraska, Kansas, Oklahoma and Texas with the potential to have converted native sod to crop. High resolution USDA NAIP imagery was combined with estimated changes to and from cropland using the CDL and NLCD to examine locations. A visual assessment was performed and many areas of change to and from cropland were identified. Landsat satellites in LandTrendr software tools allowed further analysis of potential change to and from cropland since 1985 in these locations. The results indicated a similar number of hectares changing to cropland (7901) as those going into grassland in these counties (6145).

According to USDA data, national crop net profits declined to the negative while harvested acres in hay increased in the late 1990s and early 2000s in these counties. This most likely led to a decline in crop acres planted with these lands being idle. Six years later, when corn and soybean net values went back up as a result of biofuels, hay acres declined indicating a strong likelihood that land brought into production in 2007/2008 was most likely land previously in crops and not native land. Recent yield improvements have also reduced the need for more cropland, and net profits for corn and soybeans were close to zero in 2018, 2019, and 2020 supporting the idea that corn and soybean may be over-supplied. These results better match USDA crop data, which showed few changes in overall crop hectares in recent years.

The results from this study indicate that government agencies should take previous land use, pricing, and acreage into account when analyzing land cover/land use change to crop rather than treat the beginning of the introduction of biofuels as an event that impacted a static landscape, as the crop landscape is dynamic.

Future research should focus on combining the LandTrendr algorithms with NAIP imagery to develop a national year to year change dataset for the USA with the focus on conversion between agriculture and grassland. Higher resolution satellite imagery and much faster processing and analysis is available, and Google Earth allows for rapid analysis. This data was not available in 2007/2008, but it could help with understanding changes in more recent years, which could assist in understanding earlier trends.

**Funding:** This research was funded by the United States Soybean Export Council, grant number 22CR03U14A.

**Institutional Review Board Statement:** Not applicable.

**Informed Consent Statement:** Not applicable.

**Data Availability Statement:** All datasets for this article are publicly available with links in the references.

**Acknowledgments:** The author would like to thank Steffen Mueller from the University of Illinois at Chicago, Abby Rinne and Tarik Eluri from the United States Soybean Export Council, and Edelyn Verona from the University of South Florida for reviewing the article.

**Conflicts of Interest:** The author declares there are no conflict of interest.

## Glossary

| | |
|---|---|
| CDL | USDA's Cropland Data Layer |
| NLCD | National Land Cover Database |
| CRP | Conservation Reserve Program |
| USDA | United States Department of Agriculture |
| NASS | The USDA's National Agricultural Statistics Service |
| USA | United States of America |
| NRI | USDA's Natural Resources Inventory |
| FSA | USDA's Farm Services Agency |
| Census | USDA's Census of Agriculture |
| NAIP | USDA National Aerial Imagery Program |
| MLRC | Multi-Resolution Land Characteristics Consortium |

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
