# Peer review of "Combining Tabular and Satellite-Based Datasets to Better Understand Cropland Change"

_land, doi:10.3390/land11050714_

Round 1

Reviewer 1 Report

  1. Re-select keywords. I recommend listing only 3-5 keywords.
  2. Lines 34, 97, 112, 224, 241, 267, and 364 all have the "Error! reference source not found" problem.
  3. The Y-axis of Figure 1, Figure 3, and Figure 9-12 are all missing units.
  4. Figure 1 does not indicate where used in this manuscript.
  5. Line 193, I suggest numbering the "USDA National Aerial Imagery Program (NAIP)" as "2.1.6".
  6. Please delete the ":" in "3. Methods:", and suggest adding the "Result" part to make the paper organization more reasonable.
  7. Some fonts in lines 350 and 353 are bolded, please unify the font style of the paper.
  8. The sources of data sets are different, resulting in inconsistent data standards, which will bring errors to the analysis results. I suggest that the authors add limitations to the use of different data sets in this paper.
  9. On the whole, the paper is more like a popular science paper, which is not very professional.

Reviewer 2 Report

The structure of this article is inadequate. For example, citation is not good, section names are inadequate and so on.

In addition to this, there is no objective in the introduction and it is not clear what the author want to convey.

I think the authors should reconstruct and rewrite whole the article.

Reviewer 3 Report

The author uses satellite and statistical data to examine the changes in cropland in the United States. The strength of the manuscript lies in the methodology, although it uses well-established solutions (e.g., similar to the solution implemented in LandTrendr was used to identify field crops for primary checks of agricultural subsidy claims in Hungary based on IRS satellite imagery in the early 2000s). The description of both the databases used and the methodology is adequate and the results of the research are likely to be replicable. The structure of the manuscript is also logical, but the content of the individual sections does not always meet expectations.

Abstract: there is no statement of the problem and why the author is researching this topic using the example of the USA, the presentation of the results is not adequate, I suggest that two sentences from the conclusions be included here.

Introduction: the author starts the paper in medias res, it would be good to have a paragraph about the problem the paper will address. At the end of the introduction, it would be useful to formulate the research questions and objectives and why this topic is important to study in the United States. Without these, the reader will only understand why this research was important when reading the conclusion. This is the biggest problem with the manuscript, that it presents a case study, but we don't know the background to it, why it is important, and finally, what the conclusions might be for an international readership. This is partly covered in the Conclusions (e.g.: problems with the use of moderate resolution satellite imagery) but it would be useful to highlight these more.

These changes are not significant as the materials and method and result sections are well written.

A further problem is that not all the graphs indicate the units of measurement of the data, the images (maps) are low resolution and the data sources are indicated in the titles but the author/creator is not identified.

I recommend accepting the manuscript after revision.

Additional comments:

Fig. 1. – there is no unit of measurement

Fig. 2. – I recommend reversed coloring, where red colors show declining and greens increasing cropland area, a higher resolution map would be more appropriate, and we don’t know who edited the map (own elaboration or not)

Fig. 4. – low-resolution map, I recommend reverse coloring, and we don’t know the creator of the map

Fig 5., 6. and 7. – low-resolution maps, we don’t know the creator of the maps

Reviewer 4 Report

Based on the combination of statistical data and satellite images, this paper proposes a change location method for accurately locating the inflow and outflow of cultivated land, which enables us to better understand the change of cultivated land use.  In the current era of geographic big data flooding, there are often precision differences and mismatches among data. How to effectively use these data to mine valuable association information is a very challenging and meaningful work.   

Generally speaking, the writing and structure arrangement of this paper is quite good.  I have the following suggestions for the author's reference: 

1) The title is too big. This paper only studies the transfer of cultivated Land, so it is not suitable to directly talk about Land cover.  If necessary, a case of Cropland study should be included; 

2) It is better to attach a list of professional terms at the end for easy reading, because there are many abbreviations in this article, no less than 10 as far as MY statistics are concerned; 

3) Draw an overall flow chart in the Methods section to facilitate readers to understand the idea of location; 

4) All illustrations and tables should be quoted in the text; 

5) The resolution of some images is too low to see the font clearly, such as Figure 9. 

Round 2

Reviewer 1 Report

The revised manuscript has improved quality compared to the previous version, but there are still some problems, which I marked in the manuscript in yellow.

Fig.5 I don't understand what the author means?

Author Response

Hello Reviewer,

Thank you for taking the time to review the paper a second time.  I added significantly to the literature review and the discussion to better explain the paper's background, purpose and hopefully its importance to science.  I also tried to re-word figure five so that it was more clear.  I hope you see these as positive changes and welcome any additional review you may have.

Ken Copenhaver

Round 3

Reviewer 1 Report

I think the current version can be accepted for publication.